# Comparison of the Efficacy of Different Techniques for the Removal of Root Canal Filling Material in Artificial Teeth: A Micro-Computed Tomography Study

**DOI:** 10.3390/jcm8070984

**Published:** 2019-07-07

**Authors:** Tuan Anh Nguyen, Yaelim Kim, Euiseong Kim, Su-Jung Shin, Sunil Kim

**Affiliations:** 1Microscope Center, Department of Conservative Dentistry and Oral Science Research Center, Yonsei University College of Dentistry, 50-1 Yonsei-Ro, Seodaemun-Gu, Seoul 03722, Korea; 2Department of Electrical & Electronic Engineering, Yonsei University College of Engineering, 50 Yonsei-Ro, Seodaemun-Gu, Seoul 03722, Korea; 3Department of Conservative Dentistry, Gangnam Severance Dental Hospital, Yonsei University College of Dentistry, 211 Eonju-Ro, Gangnam-Gu, Seoul 06273, Korea

**Keywords:** endodontic retreatment, root canal retreatment, Gentlefile Brush, passive ultrasonic irrigation, micro-CT

## Abstract

This study aimed to assess the efficacy of canal filling material removal using three different techniques after filling with a Gutta–Percha (GP) cone and calcium silicate-based sealer, by measuring the percentage of volume debris of GP and sealer remaining intracanal with micro computed tomography (micro-CT). The filling material was removed from 30 plastic teeth by a nickel–titanium (Ni–Ti) rotary retreatment system. Final irrigation was performed with 2 mL of saline and 10 specimens were randomly allocated to a conventional group. In the passive ultrasonic irrigation (PUI) group, ultrasonic irrigation was added to the conventional group (*n* = 10). In the Gentlefile Brush (GF Brush) group, irrigation with GF Brush was added to the conventional group (*n* = 10). Remaining filling material was measured using micro-CT imaging analysis. The total mean volume of residual filling material after retreatment in the conventional group, PUI group and GF Brush group were 4.84896 mm^3^, 0.80702 mm^3^, and 0.05248 mm^3^, respectively. The percentage of filling material remaining intracanal was 6.76% in the conventional group, 1.12% in the PUI group and 0.07% in the GF Brush group. This study shows that the cleaning effect of the GF Brush system is superior to those of Ni–Ti retreatment files and the PUI system in the apical area.

## 1. Introduction

The goal of endodontic treatment is the eradication of harmful microorganisms from the root canal. Thus, cleaning and shaping are key for the success of endodontic treatment. However, the anatomical complexities of the root canal system and limitations in current preparation and irrigation techniques lower the success rates for endodontic treatment. Studies concerning the morphology of the root canal system have shown wide variances in the canal shape and the presence of two or more canals in a single root. Furthermore, complete disinfection in the presence of several curvatures and narrow canals is difficult to achieve by all known techniques, whether chemical or mechanical. Consequently, the reported success rate for root canal treatment (RCT) is approximately 75% [1]. Although RCT is a reliable and highly successful treatment, some cases do exhibit post-treatment disease.

Nonsurgical RCT is the first option for the treatment of postendodontic disease. The retreatment procedure is mostly similar to the initial RCT procedure, with the greatest difference being the removal procedure for the root canal filling material during retreatment. There could be some necrotic tissue or bacteria among the filling material, which potentially cause persistent inflammation and pain. Therefore, dental materials in the root canal system should be completely removed in the initial step of retreatment. However, filling materials such as gutta-percha (GP) and sealers are difficult to remove because they are trapped within the irregular root canal system.

Several techniques have been used for the removal of filling material from root canals; these include stainless steel (SS) hand files, nickel–titanium (Ni-Ti) rotary instruments, and ultrasonic tips [2,3]. Rotary instruments are widely used and reportedly remove filling material in a safe and efficient manner, with high success rates [4,5]. Nevertheless, studies have shown that none of these retreatment procedures can completely clean the root canal wall, particularly in the apical third [6,7]. Selection of an instrument that can effectively clean the GP and sealer debris in the apical third of root canals is very important, considering most instruments are generally interrupted by the various curves within the canal system. In addition to rotary instruments, ultrasonic instruments are used as auxiliary tools for cleaning root canals. Passive ultrasonic irrigation (PUI) has the potential to remove dentinal debris, organic tissue, and calcium hydroxide from inaccessible root canal areas [8,9]. Grischke et al. reported that an ultrasonic irrigation protocol was superior to other techniques investigated for the removal of sealer from the root canal surface during endodontic retreatment [10]. In other studies, the use of PUI after mechanical instrumentation ensured more efficient material removal during endodontic retreatment than did other techniques such as chloroform irrigation, xylene irrigation, and eucalyptol irrigation [11,12].

Recently, a new SS system known as Gentlefile (GF; MedicNRG, Kibbutz Afikim, Israel) was released. Even though the instruments are made of SS, they have shown better mechanical properties relative to those of Ni–Ti instruments. Moreinos et al. reported that the GF system required longer time and more rotations to fracture compared with the ProTaper and RevoS systems, and the GF system applied less vertical force to the canal in comparison with the ProTaper and RevoS systems [13]. The GF system also offers a brush (GF Brush) comprising six SS strands that automatically open outwards when operated in a handpiece with a speed of 6500 rpm. The original aim of the GF Brush is to aid irrigation, but it is expected to show excellent efficiency for the removal of substances from root canals because of its design. Neelakantan et al. examined the effectiveness of irrigant agitation with the GF Brush after root canal preparation and concluded that the use of the GF Brush resulted in significantly less pulp tissue remnant compared with syringe irrigation [14]. A combination of flexibility and centrifugal movement would facilitate access and cleaning in irregular parts. In case of retreatment, the GF Brush is expected to aid in the removal of GP and sealer particles stuck on the canal walls. Although there is a study on the efficacy of the GF Brush in initial root canal treatment, no studies have compared the removal efficiency of canal filling material using the GF Brush.

The aim of this study was to assess the efficacy of canal filling material removal using three different techniques after filling with a GP cone and calcium silicate-based sealer, by measuring the percentage of volume debris of GP and sealer remaining intracanal with micro computed tomography (micro-CT). The null hypothesis was that the GF Brush system would demonstrate similar efficacy in removal of canal filling material as the retreatment Ni–Ti file and PUI system.

## 2. Material and Methods

### 2.1. Preparation of Tooth Samples

Thirty-three artificial teeth made of plastic (TrueTooth, Dental Cadre, Santa Barbara, CA, USA) were used for this study (Figure 1a). The teeth were customized samples reproducing the shape of the human mandibular first premolar and exhibiting access openings with a type I canal as per Weine’s classification [15]. A #15 K-file (Dentsply Maillefer, Ballaigues, Switzerland) was inserted into the canal to determine the working length (WL), which was recorded as 21 mm. A #40 master apical file was used. The canal in all samples was instrumented using the ProTaper Next Ni–Ti system (Dentsply Maillefer) coupled with the Dentsply X-Smart Plus motor (Dentsply Maillefer). According to the manufacturer’s instructions, X1, X2, and X3 files were used up to the full WL. Between instruments, each canal was irrigated using distilled water via a 27-gauge needle (Korean Vaccine Co., Seoul, Korea). After the instrumentation was complete, all canals were dried with #25 paper points (Dentsply Maillefer). Subsequently, three teeth were randomly selected for micro-CT, and the acquired images were overlapped for the confirmation of consistency in the prepared canal space. The other 30 teeth were prepared for obturation.

Obturation was performed using the single-cone technique. The canals were first coated with a calcium silicate-based sealer (Well-Root ST sealer, Vericom, Chuncheon-si, Korea) via a 24-gauge needle tip provided by the manufacturer. The tip was slowly pulled toward the orifice from the point of engagement in the canal. Then, medium to large GP cones (DiaDent, Cheongju-si, Korea) were customized to size 40 using a GP gauge (Dentsply Maillefer), and a single cone was inserted in each canal. The cone was gently moved with an up–down motion three times to facilitate better penetration of the sealer into finer structures, following which it was cut at the orifice level using System B (SybroEndo, Orange, CA, USA) and vertically condensed. The access cavities were filled with Caviton (GC Corporation, Tokyo, Japan). All filled samples were stored in a humidified chamber (Changshin Science, Seoul, Korea) at 100% relative humidity and 37 °C for 7 days until retreatment. All procedures were performed by a single operator.

### 2.2. Root Canal Retreatment

The temporary filling was removed with a round bur, and the specimens were randomly allocated to three different groups (*n* = 10 per group) according to the material removal technique.

#### 2.2.1. Conventional Group

The root canal filling material was removed using ProTaper Universal retreatment files (Dentsply Maillefer) according to the manufacturer’s instructions. D1, D2, and D3 files were sequentially used with the crown-down technique until WL was reached. The files were manipulated with a brushing action at a constant speed of 500 rpm, as recommended. A solvent was not used. Retreatment was considered complete when no GP/sealer was visible on the surface of the instruments. Root canal refinement was accomplished using #25, 30, 35 and 40 K-files up to WL. Between instruments, the canal was irrigated with 2 mL of distilled water via a 27-gauge needle (Korean Vaccine Co.). Final irrigation was performed with 5 mL of distilled water for 30 s. Finally, the root canal was dried with paper points and stored in a dry environment for micro-CT analysis.

#### 2.2.2. PUI Group

In the PUI group, ultrasonic irrigation was added to the procedure described for the conventional group. Ultrasonic irrigation was performed using an ultrasonic endodontic tip (Endosonic Blue, Maruchi, Chuncheon-si, Korea; Figure 1b), which was inserted into the root canal up to 1 mm short of WL and oscillated toward the apex [16]. Activation with 3 mL of distilled water for 60 s was performed three times (total 3 min per tooth). The distilled water was replenished between each activation cycle.

#### 2.2.3. GF Brush Group

In the GF Brush group, irrigation with the GF Brush was added to the procedure described for the conventional group. Final instrumentation was performed using the GF Brush (Figure 1c,d), which was inserted into the root canal up to 1 mm short of WL. Activation with 3 mL of distilled water for 60 s was performed three times (total 3 min per tooth). The distilled water was replenished between each activation cycle.

### 2.3. Micro-CT Analysis and Stereomicroscopy

Micro-CT and image analysis were performed as previously described [17]. A high-resolution micro-CT scanner (SkyScan 1173, Bruker, Billerica, MA, USA) was used to scan the samples in the three groups. All acquired images were reconstructed using NRecon software, version 1.6.6.0 (Bruker microCT, Kontich, Belgium). For evaluation of the residual filling material, three-dimensional (3D) images of the filling material were visualized by surface-CT-Vol (SkyScan). The CT-An software (SkyScan) was used to measure the volume of the prepared canal space in the three sample specimens and the volume of the residual filling material after retreatment in the remaining 30 specimens. The apical region was defined as the area between 1 and 5 mm from the apex, the middle region as the area between 5 and 10 mm from the apex, and the coronal region as the area between 10 and 15 mm from the apex.

After micro-CT, the teeth were longitudinally sectioned at the labial and lingual surfaces using a low-speed diamond wheel (Struers Minitom, DK-2610, Rodovre, Denmark) under water cooling. Each root surface was observed under a stereomicroscope (Zeiss, Gottingen, Germany).

### 2.4. Statistical Analysis

The amount of residual filling material was expressed as a percentage of the total area of each section in the root canal. The measurements were evaluated by a single observer blinded to the study groups. The Shapiro–Wilk test was used to verify whether the data were normally distributed. The Student’s *t* test was used to compare the percentage volume of residual filling material in the apical, middle, and coronal regions. A *p*-value of <0.05 was considered statistically significant. All statistical analyses were performed using SPSS software, version 23 (SPSS, Chicago, IL, USA).

## 3. Results

The volume of the prepared canal space was similar in the three sample specimens (71.68401 mm^3^). The total mean volume of residual filling material after retreatment in the conventional, PUI, and GF Brush groups was 4.84896, 0.80702, and 0.05248 mm^3^, respectively, with percentage values of 6.76%, 1.13%, and 0.07%, respectively (Table 1; Figure 2). In the conventional group, the filling material was distributed evenly on the root canal walls. In the PUI group, the filling material debris was mostly concentrated in the apical region, which was beyond the canal curvature. In the GF Brush group, all three regions showed a small amount of residual material.

Stereomicroscopic observation demonstrated that residual filling material stayed on the root canal walls. In the conventional group, the material was uniformly distributed in the three regions, while the PUI and GF Brush groups exhibited an insignificant amount of residual material. In the PUI group, most of the residual material was concentrated in the apical region (Figure 3).

## 4. Discussion

In the present study, we compared the efficacy of filling material removal during retreatment in artificial mandibular premolars between the GF Brush, conventional Ni–Ti, and PUI systems. The reproduction of the clinical situation may be regarded as the major advantage of the use of natural teeth for experiment. However, the wide range of variations in three-dimensional root canal morphology makes standardization difficult between groups. On the other hand, the use of artificial teeth allows standardization of degree, location, and radius of root canal curvature in three-dimensions [18]. Thus, artificial teeth were selected to obtain more reliable results in terms of the volume of residual filling material after retreatment. Although uniform samples were used, the prepared canal shapes may have differed. Therefore, to determine consistency, we randomly selected three prepared teeth and subjected them to micro-CT. The acquired images were superimposed, and the results showed negligible variations in the prepared canal space among the teeth. This was probably because the canal space in artificial teeth is quite wide, even before canal preparation. Larger files were not used because we were not focused on comparing the preparation efficiency of the instruments. Consequently, we could maintain a consistent canal space volume in all specimens.

We found that the amount of residual filling material was smaller in the PUI group than in the conventional group, with the exception of the apical region. This result was similar to that of a previous study, where the PUI technique was found to be more effective than the conventional technique for the removal of root canal filling material from the cervical and middle thirds during endodontic retreatment [19]. Additional PUI retreatment eliminates sealer and GP debris beyond the curvature or in areas that cannot be accessed by conventional retreatment files. During PUI, free intracanal movement of the file is necessary for easy penetration of the solution into the root canal system and a more powerful cleaning effect [20]. During PUI, energy is transmitted from a file or smooth oscillating wire to the irrigant by means of ultrasonic waves that induce two physical phenomena: stream and cavitation of the irrigant. The acoustic stream can be defined as rapid movement of the fluid in a circular or vortex shape around the vibrating file, while cavitation is defined as the generation of steam bubbles or the expansion, contraction, and/or distortion of pre-existing bubbles in a liquid [21]. The effect of an ultrasound tip is the maximum in the middle and coronal regions because of the direction of operation [8]. In our samples, the apical region was located beyond curvature from the canal orifice. This may have caused limited movement of the tip during its passage through the curvature, resulting in reduced cleaning efficiency in the apical third.

Compared with the conventional and PUI groups, the GF Brush group showed a significantly smaller volume of residual filling material in the apical region. Almost 99.76% of the filling material in the apical region was eliminated by this system. The superior effects of the GF Brush system may be attributed to the mechanism of the GF Brush. When the brush is not rotating, the strands are present in a twisted form. However, at a high rotating speed, the strands open to cover the entire canal diameter. These opened strands mechanically remove the remaining debris and bring the irrigant into intimate contact with the canal surface. Flexibility and centrifugal movement facilitate access and cleaning in irregular parts. These characteristics of the GF Brush system may contribute to the better cleaning efficiency throughout the canal system, particularly the apical region. To achieve more effective cleaning and shaping of the root canal system, the endodontist must ensure cleanliness of the apical third of the canal, and we believe that the GF Brush system is an effective tool to achieve this goal.

This study has some limitations. First, the artificial teeth used in this study cannot perfectly replicate natural teeth due to complex root canal anatomy such as lateral canal, isthmus, and fin. In addition, the microscopic structures of the dentine of natural teeth are absent in the artificial teeth; therefore, the adhesion between the endodontic filling material and the root canal wall cannot be reproduced.

Within the limitation of this in vitro study, our findings suggest that the GF Brush system is superior to conventional Ni–Ti retreatment files and PUI in terms of effective removal of root canal filling material during retreatment, particularly from the apical third of the canal. This is because the cleaning effect of the GF Brush is not impeded by the curvature of the canal because of its design. Additional experiments in various shapes of root canal and/or natural teeth may be helpful in clinical applications.

## Figures and Tables

**Figure 1 jcm-08-00984-f001:**
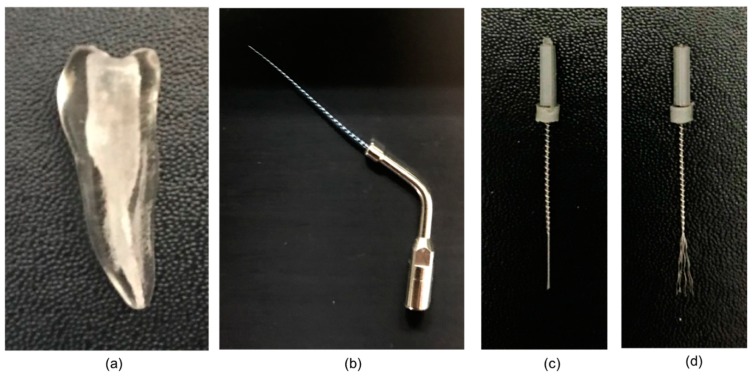
(**a**) Artificial plastic tooth sample; (**b**) Endosonic Blue tip; (**c**) strand twisted Gentlefile Brush; (**d**) strand unfolded Gentlefile Brush.

**Figure 2 jcm-08-00984-f002:**
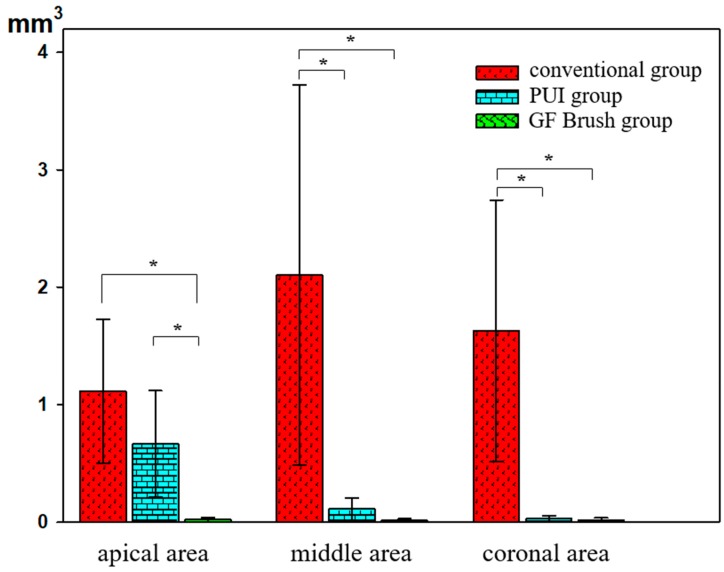
Volume of remaining filling materials in three different groups. ’*’ represents statistically significant differences between the groups within the same area (*p* < 0.05). PUI, Passive ultrasonic irrigation; GF Brush, Gentlefile Brush.

**Figure 3 jcm-08-00984-f003:**
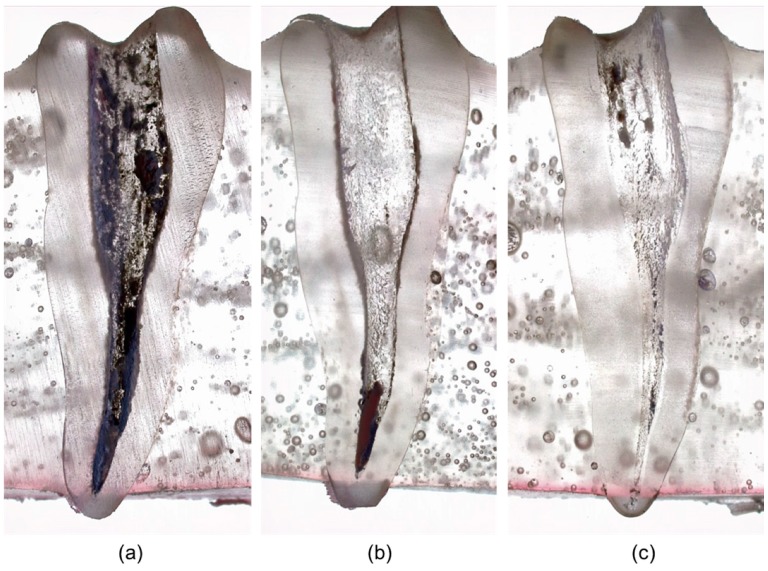
Stereomicroscope images of remaining intracanal filling materials in three groups: (**a**) conventional group; (**b**) passive ultrasonic irrigation group; (**c**) Gentlefile Brush group.

**Table 1 jcm-08-00984-t001:** Volume and percentage of remaining filling materials in three different groups.

Group		Apical Area	Middle Area	Coronal Area	Total
Conventional group	Vm (mm^3^)	1.11580 ± 0.61216	2.10583 ± 1.61961	1.62733 ± 1.11357	4.84896 ± 2.83251
	%Vm (%)	13.33 ± 7.32	18.06 ± 13.89	3.15 ± 2.16	6.76 ± 3.95
PUI group	Vm (mm^3^)	0.66543 ± 0.45265	0.11165 ± 0.09662	0.02994 ± 0.02576	0.80702 ± 0.50795
	%Vm (%)	7.95 ± 5.41	0.96 ± 0.83	0.06 ± 0.05	1.13 ± 0.71
GF Brush group	Vm (mm^3^)	0.02024 ± 0.02070	0.01567 ± 0.01308	0.01657 ± 0.01975	0.05248 ± 0.04562
	%Vm (%)	0.24 ± 0.25	0.13 ± 0.11	0.03 ± 0.04	0.07 ± 0.06

Vm: Volume of filling materials remaining inside the root canal; %Vm: Percentage of volume of filling materials remaining inside the root canal. PUI, Passive ultrasonic irrigation; GF Brush, Gentlefile Brush.

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
