# Peer review of "Comparison of the Efficacy of Different Techniques for the Removal of Root Canal Filling Material in Artificial Teeth: A Micro-Computed Tomography Study"

_jcm, 2019, doi:10.3390/jcm8070984_

Round 1
Reviewer 1 Report
The authors have discreetly described the study they conducted. However, it is advisable to reinforce the conclusions in light of the results and evidence in the scientific literature.
Introduction
1. Add bibliographic references of the studies in the literature concerning the Gentlefiles. 2. Make a brief summary of the literature studies on gentlefiles 3. Specify in what the present study differs from the study conducted by Neelakantan P et al 2018 internationa endodontic journal.
Materials and methods· Specify better the reason for choosing "artificial teeth made of plastic" instead of natural teeth.
· It is my opinion that bringing the ultrasonic tip in the PUI technique to 1 mm from the WL increases the risk of extrusion of the canal irrigant beyond “in vivo” apex.justify this choice with bibliographical references.
Discussion
Line 192 “We used the ProTaper X3 file as the final file, and this file is commonly used for mandibular premolars in clinical practice as well” add bibbliografia
Author Response
The authors have discreetly described the study they conducted. However, it is advisable to reinforce the conclusions in light of the results and evidence in the scientific literature.
-> Thank you for your advice. We have added a reference that can support our findings and reinforce the conclusion. The added parts are marked underline.
Introduction
1. Add bibliographic references of the studies in the literature concerning the Gentlefiles.
-> We have added the following article as a reference in accordance with your advice. (line 78-80)
Neelakantan, P.; Khan, K.; Li, K.Y.; Shetty, H.; Xi, W. Effectiveness of supplementary irrigant agitation with the Finisher GF Brush on the debridement of oval root canals instrumented with the Gentlefile or nickel titanium rotary instruments. Int Endod J 2018, 51, 800-807, doi:10.1111/iej.12892.
2. Make a brief summary of the literature studies on gentlefiles
-> We have added a brief summary of the studies related to gentlefiles to the introduction. (Line 72~83)
3. Specify in what the present study differs from the study conducted by Neelakantan P et al 2018 international endodontic journal.
-> We have added the difference between Neelakantan's study and our study to the introduction. (Line 78~83)
Materials and methods
Specify better the reason for choosing "artificial teeth made of plastic" instead of natural teeth.
-> We have elaborated on the reason for choosing artificial teeth instead of natural teeth in the first paragraph of the discussion. (Line 189~194) Also, we have added the limitations of using artificial teeth for our experiment to the last part of the discussion. (Line 230~240)
It is my opinion that bringing the ultrasonic tip in the PUI technique to 1 mm from the WL increases the risk of extrusion of the canal irrigant beyond “in vivo” apex.justify this choice with bibliographical references.
-> Thank you for your comment. As your opinion, the possibility of extrusion would increase when the tip is inserted deeply to the apex. However, according to a recent systematic review of ultrasonic irrigation, ultrasonic files/tips were inserted 1 mm shorter than the working length in most studies. We added the following article as a reference. (Line 136)
Caputa, P.E.; Retsas, A.; Kuijk, L.; Chavez de Paz, L.E.; Boutsioukis, C. Ultrasonic Irrigant Activation during Root Canal Treatment: A Systematic Review. J Endod 2019, 45, 31-44.e13, doi:10.1016/j.joen.2018.09.010.
Discussion
Line 192 “We used the ProTaper X3 file as the final file, and this file is commonly used for mandibular premolars in clinical practice as well” add bibbliografia
-> This sentence describes the technique used in our clinic, not having scientific evidence. We have deleted this sentence.
Reviewer 2 Report
The article title and abstract are appropriate, however, I think authors should specify that is an in vitro study. The purpose of the article and its significance is stated clearly and the writing is clear and concise.
However, I think that results and conclusions, although interesting, are preliminary because they cannot be completely translated to clinical activity. This in vitro study has been performed on artificial teeth made of plastic, so the microscopical structure of these samples cannot be compared to the endodontic walls of natural teeth, so there could be a risk of bias, that could affect the adhesion of the endodontic filling material. For this reason, also conclusions should be taken with care, because of the risk of bias, intrinsic to the chosen of the samples.
I think that the paper is well structured and written; however, considering the high impact factor of the journal of clinical medicine, results are too preliminary to be considered of interest to members of the education research community.
Author Response
The article title and abstract are appropriate, however, I think authors should specify that is an in vitro study. The purpose of the article and its significance is stated clearly and the writing is clear and concise.
-> Thank you for your comments. We have added 'in Artificial Teeth' in the title to help readers understand as your advice.
However, I think that results and conclusions, although interesting, are preliminary because they cannot be completely translated to clinical activity. This in vitro study has been performed on artificial teeth made of plastic, so the microscopical structure of these samples cannot be compared to the endodontic walls of natural teeth, so there could be a risk of bias, that could affect the adhesion of the endodontic filling material. For this reason, also conclusions should be taken with care, because of the risk of bias, intrinsic to the chosen of the samples.
-> Thank you for your helpful comments. We agree with your opinion that the artificial teeth can't perfectly replicate the natural teeth and therefore, there could be a risk of bias in result. However, as we have mentioned in the discussion, we also considered variables due to wide range of different tooth morphology in natural teeth. We had a thorough discussion about this concern when designing experiments and we have concluded that the unification of variables such as root curvature, root length, isthmus, etc. is more important in comparing the removal efficiency of filling material between the groups. There is an advantage to using an artificial tooth instead of natural tooth, so we do not think that the results of our study are preliminary just because we used an artificial tooth in our study.
We agree with your comments that there are limitations of using artificial tooth, so we have added the limitations of using artificial tooth in our experiment to the last part of the discussion. Also, we have elaborated on the reason for choosing artificial tooth instead of natural tooth in the first paragraph of the discussion.
I think that the paper is well structured and written; however, considering the high impact factor of the journal of clinical medicine, results are too preliminary to be considered of interest to members of the education research community.
-> Removal of previous filling material in the root canal is very important in retreatment of infected cases. However, complete removal of canal filling materials with existing equipment is almost impossible. Recently released Gentlefile Brush was significantly more efficient in removal of gutta percha compared with other methods used. This result indicate that supplementary irrigant agitation with Gentlefile Brush may allow improved disinfection in retreatment; therefore, we think our research has clinical significance. Based on the results and conclusion of our study, we believe our paper is appropriate to be reviewed in a special issue of the Journal of Clinical Medicine, 'Innovations in Endodontic Dentistry'.
Round 2
Reviewer 2 Report
The paper has been improved and is now acceptable